# Effects of interactivity, immersion, and physical discomfort on learning in VR nursing education

**Yu-Chia Chang[1], Yi-Ting Lo[2], Cheng-Chia Yang [2]\***

**1** Department of Long Term Care, National Quemoy University, Jinning, Kinmen County, Taiwan,
**2** Department of Healthcare Administration, Asia University, Wufeng, Taichung, Taiwan

\* chengchia@asia.edu.tw

## Abstract

### Background

Although VR provides innovative opportunities for nursing education, little is known about how its core features immersion and interactivity shape learners' self-efficacy, motivation, and cognitive processes. Furthermore, the impact of physical discomfort during VR use remains underexplored. This study investigates the effects of immersion and interactivity on affective and cognitive constructs in nursing education using virtual reality (VR), while examining the influence of physical discomfort and the moderating roles of age and gender.

### Methods

A cross-sectional study was conducted with 209 students (aged ≥20 years) recruited from three nurse aide training centers in Taiwan. The research team developed a VR training system using Unity 3D and Autodesk 3DS Max, incorporating three essential nursing skills: Heimlich maneuver, meal preparation/feeding/medication assistance, and vital sign measurement. After completing the VR training, participants filled out structured questionnaires. Data were analyzed using partial least squares structural equation modeling (PLS-SEM).

### Results

Interactivity showed stronger positive associations than immersion with self-efficacy, intrinsic motivation, situational interest, and embodied learning, and was associated with lower levels of extraneous cognitive load. Visual discomfort was negatively associated with both immersion and interactivity, whereas neck, shoulder, and back (NSB) pain was positively associated with immersion. Multi-group analysis revealed that female participants reported greater visual discomfort, which reduced immersion and interactivity. Additionally, participants under 35 years old exhibited greater sensitivity to visual discomfort in relation to immersion compared with older participants.

**Data availability statement:** The data underlying the findings of this study contain sensitive human participant information and therefore cannot be made publicly available due to ethical restrictions imposed by the Institutional Review Board of Taichung Jen-Ai Hospital (Approval No. 110–82). In accordance with the IRB-approved protocol, de-identified data may be made available to qualified researchers upon reasonable request and with approval from the Institutional Review Board of Taichung Jen-Ai Hospital. Data requests should be submitted to the Institutional Review Board of Taichung Jen-Ai Hospital (Email: jahirb@mail.jah.org.tw).

**Funding:** The author(s) received no specific funding for this work.

**Competing interests:** The authors have declared that no competing interests exist.

## Conclusions

Interactivity is more crucial than immersion for enhancing affective and cognitive learning outcomes in VR-based nursing education. Visual discomfort significantly impairs learning experiences, with stronger negative effects among female and younger users, suggesting the need for further attention to user characteristics and physical comfort in VR-supported nursing education.

---

## 1 Introduction

Numerous information technologies, such as those related to e-learning, have been applied in nursing education, resulting in the higher learning motivation of nursing students and making learning processes more engaging and effective [1]. Although e-learning systems have been praised for optimizing learning and teaching processes by providing a convenient and flexible approach that allows learning anytime and anywhere, they still face criticism for lacking sufficient interactivity and appeal [2]. In comparison, virtual reality (VR)–based learning systems retain the advantages of e-learning while adding immersive and interactive features that enhance student motivation and allow the safe simulation of high-risk clinical scenarios, thereby minimizing students' exposure to potential harm or safety risks during real-life training [3,4].

In nursing education, prior studies have shown that VR-based instruction is associated with higher satisfaction, greater knowledge gains, improved skill performance, and increased self-efficacy [5]. Previous studies have further identified several psychological processes through which VR may support learning, including positive associations with intrinsic motivation, situational interest, and self-efficacy, alongside reduced extraneous cognitive load [6]. VR environments also allow learners to revisit clinical scenarios repeatedly and to make real-time decisions in safe, simulated settings, facilitating reflection and the development of clinical judgment [7]. Such features are particularly valuable for mastering procedural skills that require repeated practice and hands-on experience [5,7–11]. Accordingly, VR technologies have become an important instructional approach for supporting learning in nursing education [12,13].

While VR applications in nursing education have been extensively researched, most studies have not fully explored how the unique characteristics of VR, particularly immersion and interactivity, contribute to the learning process [14]. Immersion refers to the sensory fidelity of the virtual environment and the extent to which users feel perceptually surrounded by it [15,16]. Interactivity concerns the degree to which users can manipulate or respond to virtual elements and observe the outcomes of their actions [16]. The two key technologies decide the fidelity of situations perceived by learners and the degree of independence in controlling learning felt by learners [17]. Recent studies have suggested that these features may influence learners' emotional and cognitive experiences, including intrinsic motivation, situational interest, embodied learning, and self-efficacy, while also potentially increasing cognitive load

due to the realism and interactivity of the environment [3]. Despite these insights, how these VR affordances relate specifically to learning mechanisms in nursing education remains insufficiently understood.

VR use can also elicit various forms of physical discomfort, including nausea, dizziness, eyestrain, and musculoskeletal strain [18–21]. These symptoms are often attributed to sensory–vestibular conflict, visual fatigue, and repeated static or dynamic postural demands imposed by head-mounted displays [22,23]. Such discomfort has been shown to reduce users' engagement and willingness to continue VR-based activities [24]. Prior research further indicates that susceptibility to VR-induced discomfort varies across individuals, with age- and gender-related differences observed in visual strain and cybersickness sensitivity [25,26]. Despite these findings, relatively little is known about how different types of physical discomfort relate to immersive and interactive learning experiences in nursing education. Addressing this gap is essential for understanding how discomfort may shape learners' psychological responses in VR-based training.

In summary, this study had three objectives. First, it examined how immersion and interactivity influence learners' intrinsic motivation, situational interest, embodied learning, extraneous cognitive load, and self-efficacy. Second, it analyzed whether specific adverse reactions generated by VR use, namely head discomfort, musculoskeletal discomfort, and visual discomfort, affect these immersive and interactive learning processes. Finally, it investigated whether individual differences such as gender and age moderate the relationship between adverse reactions and immersive and interactive cognition in VR environments. The outcomes of this study could have considerable theoretical and practical significance for facilitating VR-based learning processes in nursing education.

## 2 Theoretical background and hypotheses

### 2.1 Impacts of VR on emotions and cognition in learning

Recent systematic reviews of VR and AR applications in healthcare have highlighted immersion and interactivity as key determinants of users' experience [27]. However, much of this work remains focused on experiential outcomes rather than on how these VR affordances relate to underlying learning mechanisms such as motivation, self-efficacy, or cognitive processing. Understanding these mechanisms is essential for explaining how learners engage with and interpret VR-based instructional environments.

Immersion and interactivity are two core affordances of VR that can influence both affective factors in learning, such as intrinsic motivation, situational interest, embodied learning, self-efficacy, and cognitive factors, including the management of extraneous cognitive load [3]. These constructs are central to understanding engagement in immersive learning environments, and recent studies have reported consistent associations between VR features and these psychological responses [6].

Intrinsic motivation refers to engaging in an activity for the inherent satisfaction and enjoyment it provides, driven by the fulfillment of autonomy, competence, and relatedness needs [28]. Immersive environments can enhance relatedness and competence by situating learners in authentic, meaningful contexts [29,30], while interactivity supports autonomy and competence by enabling learners to make choices, interact with virtual elements, and receive immediate feedback [31]. Together, these features foster intrinsic motivation, which is a key driver of sustained engagement and deep learning [3,10].

Situational interest is a spontaneous emotional response to environmental stimuli, which can be both elicited and maintained through engaging learning experiences [32]. In VR contexts, immersion captures learners' attention with novel and vivid environments, whereas interactivity sustains this interest by enabling active exploration and meaningful engagement with the content [3,33].

Embodied learning deepens individuals' understanding of themselves and the surrounding world through physical perception and interaction [34]. High immersion integrates learners into realistic, action-rich contexts that align bodily experiences with learning content, supporting concrete conceptual understanding [3]. High interactivity allows learners to directly

influence the virtual environment through physical actions, linking engagement to cognitive processing and reinforcing conceptual learning [33].

Self-efficacy, defined as one's confidence in performing a task [35]. is also linked to VR affordances. Immersive VR enables learners to practice skills in realistic but low-risk environments, which may enhance perceived competence [5,13,29,36]. Interactivity allows learners to manipulate virtual elements and observe the consequences of their actions, thereby offering opportunities for mastery experiences, a central source of self-efficacy. [31,35].

Finally, Cognitive Load Theory emphasizes the limitations of working memory [37]. Immersion and interactivity can be associated with both reductions and increases in extraneous cognitive load, depending on task design. When instructional design is suboptimal, high sensory fidelity may introduce unnecessary perceptual information, while complex interaction requirements may impose cognitive demands unrelated to instructional goals [10,13,38,39]. In this context, the following hypotheses were proposed:

H1: Immersion in VR is positively associated with on intrinsic motivation (1a), situational interest (1b), embodied learning (1c), self-efficacy (1d), and extraneous cognitive load (1e).

H2: Interactivity in VR is positively associated with intrinsic motivation (2a), situational interest (2b), embodied learning (2c), self-efficacy (2d), and extraneous cognitive load (2e).

## 2.2 Impacts of physical discomfort on VR-based learning

VR use can lead to various forms of physical discomfort, including visual discomfort, limb pain, neck–shoulder–back (NSB) pain, and head discomfort. These categories have been widely used in prior research to describe users' physiological responses to immersive technology [18,40,41]. Prior VR/AR reviews in healthcare have also noted that visual and physical strain are important determinants of user experience, reflecting the physiological demands imposed by head-mounted displays and interactive environments [27].

Visual discomfort is frequently reported during VR use because head-mounted displays place sustained demands on the visual system, contributing to symptoms such as eyestrain or blurred vision [23]. Musculoskeletal pain may arise from repetitive or static postures maintained during VR interaction, particularly in the upper body [42,43]. Head discomfort, including dizziness and headache, has been associated with visually induced motion sensations commonly described in cybersickness research [44,45]. Although these reactions vary across individuals, they may influence users' perceptual engagement and comfort during VR tasks.

Because physical discomfort may compete with attentional resources or interfere with perceptual processing, it may be associated with reduced immersion or diminished ease of interaction during VR-based learning. Accordingly, this study examines whether different forms of discomfort, including visual discomfort, limb pain, NSB pain, and head discomfort, are associated with learners' perceived immersion and interactivity in VR environments.. In this context, the following hypotheses were proposed:

H3: Physical discomfort, namely head discomfort (3a), limb pain (3b), NSB pain (3c), or visual discomfort (3d), is negatively associated with immersion.

H4: Physical discomfort, namely head discomfort (4a), limb pain (4b), NSB pain (4c), or visual discomfort (4d), is negatively associated with interactivity.

## 2.3 Individual differences: The moderating effects of gender and age

Age and gender are considered the most common individual characteristics that may predict adverse reactions to VR [25,26,45,46]. Studies have indicated that women are more likely to report physical discomfort during VR use, such as nausea, dizziness, or eyestrain [45,47]. Age-related patterns have also been observed, with research indicating that susceptibility to visually or motion-induced discomfort may vary across adulthood [26]. These findings highlight the moderating role of individual differences in shaping VR experiences. In this context, people of different genders or at different

ages were considered to have different levels of physical discomfort during VR use, and the following hypotheses were proposed:

H5: Gender moderates the association between physical discomfort, namely head discomfort (5a), limb pain (5b), NSB pain (5c), or visual discomfort (5d), on immersion.

H6: Gender moderates the association between physical discomfort, namely head discomfort (6a), limb pain (6b), NSB pain (6c), or visual discomfort (6d), on interactivity.

H7: Age moderates the association between physical discomfort, namely head discomfort (7a), limb pain (7b), NSB pain (7c), or visual discomfort (7d), on immersion.

H8: Age moderates the association between physical discomfort, namely head discomfort (8a), limb pain (8b), NSB pain (8c), or visual discomfort (8d), on interactivity.

## 3 Methods

### 3.1 Samples

This study was conducted in accordance with the Declaration of Helsinki, and approved by the Institutional Review Board (or Ethics Committee) of Jen Ai Hospital, Taichung, Taiwan (protocol code IRB110−82, date of approval: 12 December 2021).. All participants were fully informed of the study objectives and procedures prior to participation. Written informed consent was obtained from each participant before data collection, in accordance with the ethical standards of the Institutional Review Board and the Declaration of Helsinki. Participants were informed of their right to withdraw from the study at any time without penalty. A cross-sectional design and quantitative survey using structured questionnaires were used, and participants were trainees aged 20 years or older, recruited from three Nurse Aide training centers in Taiwan. These training centers are professional institutions dedicated to equipping individuals with essential caregiving knowledge and practical skills required for roles in home care services and healthcare facilities. The programs provided by these centers focus on a wide range of competencies, including assisting with daily living activities, monitoring vital signs, ensuring patient safety, and providing emotional support. The exclusion criteria included individuals with neuropsychiatric, cardiovascular, cognitive, and sensory disorders, as well as those who experienced severe or debilitating discomfort during VR use that prevented them from completing the session. Data were collected between 1 June and 31 July 2022. G Power 3.1.9.7 determined a sample size of 153 based on an R-squared value of 0.15, 95% power, and 11 predictors, following the approach of Saleem et al. [48]. Of the 220 recruits, 8 withdrew due to illness (e.g., COVID-19) and 3 withdrew for personal reasons. Consequently, 209 subjects met the minimum sample size requirement.

### 3.2 Research procedures

This study used the HTC VIVE Focus Plus head-mounted display as the primary device. The curriculum followed clinical guidelines, with the study team creating the script and training scenarios. Unity 3D with C++ was used for system programming. Autodesk 3DS Max was used to create virtual backgrounds and characters. The VR curriculum focused on three essential nursing skills (Fig 1), namely, training in the Heimlich maneuver, training in meal preparation, feeding and medication assistance, and training in vital sign measurement. To ensure a uniform level of expertise among the participating students, all underwent theoretical courses on the three techniques, led by a nurse. The courses lasted 60 minutes, followed by a 10-minute break before the VR simulations began. Students completed the VR tasks at their own pace, with a 3-minute break between each task, for an average total training time of 38.9 minutes. Individual session times were 9.6 minutes for the Heimlich manoeuvre, 15.1 minutes for meal preparation, feeding, and medication assistance, and 8.2 minutes for vital signs. The training was stopped immediately if any adverse reactions were observed. After completing the VR training, participants completed an evaluation questionnaire and received a NTD 100 shopping voucher upon completion.

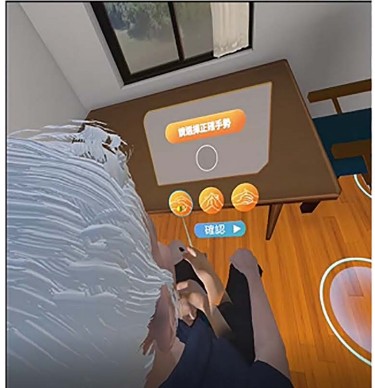 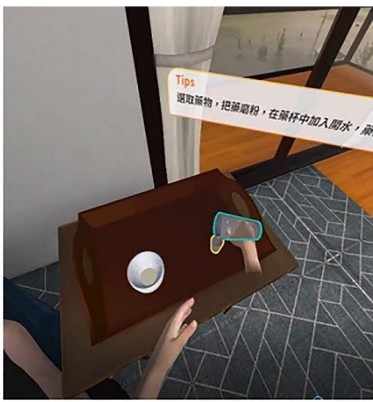 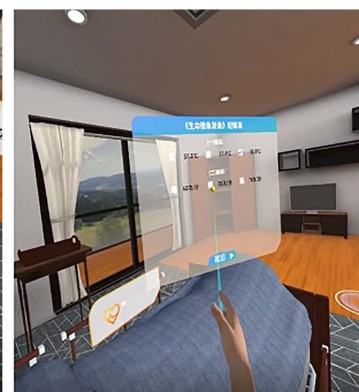

Heimlich maneuver meal preparation, feeding and medication assistance vital sign measurement

**Fig 1. Simulated displays of the VR-based teaching system used in the nursing training intervention.** The figure illustrates three representative VR learning scenarios: **(a)** Heimlich maneuver training, (b) meal preparation, feeding, and medication assistance, and (c) vital sign measurement. All scenarios were presented in an immersive virtual environment and designed to support procedural learning through interactive, first-person simulations.

To enhance reproducibility and maintain consistency across participants, each module employed a standardized sequence of actions, fixed scenario complexity, and a unified cueing system (e.g., on-screen prompts and color-change indicators). Immediate corrective feedback was provided when required procedural steps were missed or performed incorrectly. Full procedural steps, interaction rules, and cueing/feedback specifications are documented in S1 Appendix.

### 3.3 Measuring instrument of this study

The study questionnaire consisted of 43 questions in four sections. The first section assessed gender, age, education, and marital status to determine the profile of the subject. The second section assessed VR immersion and interactivity using an eight-item scale based on a previous study [49]. The third section, which focused on learning emotions and cognitive responses, consisted of 18 questions across five dimensions: intrinsic motivation (4 questions), situational interest (4 questions), embodied learning (3 questions), self-efficacy (4 questions), and extraneous cognitive load (3 questions). Embodied learning was operationalized as the extent to which learners perceived that their body movements and gestures supported their understanding and performance of the VR nursing procedures. Questions were formulated with reference to various studies [13,32,50–52]. The fourth section assessed physical discomfort after VR use, based on a scale adapted from Lin et al. [40], with 17 questions in four dimensions: head discomfort (3 questions), limb pain (5 questions), NSB pain (3 questions), and visual discomfort (6 questions). Scores ranged from 1 (strongly disagree) to 5 (strongly agree).

Because the scales were adapted for use in a VR nursing-training context, all items underwent a structured translation process. Items were first translated into Chinese by a professional translation agency. Two bilingual researchers reviewed the wording for semantic equivalence and conceptual consistency. Three experts in nursing education and VR-based training assessed item relevance and contextual appropriateness, leading to minor refinements before finalization. Complete items are provided in S2 Appendix.

### 3.4 Statistical methods

The study used SmartPLS 4, partial least squares structural equation modeling. PLS-SEM was selected because it is well suited for models involving multiple latent constructs, does not require multivariate normality, and performs robustly with moderate sample sizes. These characteristics align with the analytical needs of the present study, which includes several

psychological constructs measured with multi-item scales. Model evaluation followed established PLS-SEM guidelines [53], including assessments of measurement reliability, convergent validity, discriminant validity, and structural path estimates.

## 4 Results

### 4.1 Descriptive statistics of the samples

Table 1 summarizes sample details: 55.02% were female, 44.98% male; mean age was 35.6 years (range: 20–69). In terms of education, 61.72% were college graduates, 21.53% were high school graduates, and 13.4% had lower educational attainment. Marital status: 67.9% married, 20.1% never married, and 11.96% divorced.

### 4.2 Reliability and validity of the research instruments

In this study, goodness of fit was assessed using the standardized root mean squared residual (SRMR). If the SRMR between the saturated and estimated models was < 0.08, it indicated a strong fit; otherwise, < 0.1 indicated an acceptable fit. The SRMR in this study was 0.081, indicating a good model fit. Individual item reliability, composite reliability (CR), and average variance extracted (AVE) were used for the measurement modes. All factor loadings were > 0.5 and significant. Factor loading coefficients ranged from 0.628 to 0.958 (See Table 2).

Composite reliability (CR) of the latent variables, indicating internal consistency, ranged from 0.771 to 0.972, meeting the threshold of over 0.7 suggested by Chin [54]. The explanatory power, assessed by the average variance extracted (AVE) of the latent variables, ranged from 0.531 to 0.874, exceeding the threshold of 0.5 set by Fornell and Larcker [55] for discriminant validity. Discriminant validity, confirmed by comparing the square roots of AVE with correlation coefficients, was found in all dimensions. In summary, the analysis confirms acceptable levels of reliability, convergent validity, and discriminant validity across all dimensions (See Table 3).

### 4.3 Structural model

Fig 2 presents the standardized path coefficients. Immersion showed positive associations with situational interest ($\beta = 0.221$, $t = 2.374$, $p < 0.05$), embodied learning ($\beta = 0.261$, $t = 2.07$, $p < 0.05$) and self-efficacy ($\beta = 0.354$, $t = 3.718$,

**Table 1. Sample demographics.**

| Variable | Category | N | Valid % |
|---|---|---|---|
| Age | 20–30 | 81 | 38.76% |
| | 31–40 | 62 | 29.67% |
| | 41–50 | 38 | 18.18% |
| | 51–60 | 20 | 9.57% |
| | 60 and above | 8 | 3.83% |
| Gender | Male | 94 | 44.98% |
| | Female | 115 | 55.02% |
| Education | Less than high school | 28 | 13.40% |
| | High school graduate | 45 | 21.53% |
| | College degree | 129 | 61.72% |
| | Master | 7 | 3.35% |
| Marital status | Single | 42 | 20.10% |
| | Married/cohabiting/partnered | 142 | 67.94% |
| | Separated/Divorced/Widower | 25 | 11.96% |

**Table 2. Validity and reliability.**

| Construct | Item | Loading | T-value | CR | AVE | Cronbach's α |
|---|---|---|---|---|---|---|
| Head discomfort | HD1 | 0.886 | 13.549 | 0.946 | 0.854 | 0.917 |
| | HD2 | 0.958 | 44.956 | | | |
| | HD3 | 0.927 | 60.593 | | | |
| limb pain | LP1 | 0.923 | 45.038 | 0.972 | 0.874 | 0.93 |
| | LP2 | 0.941 | 72.098 | | | |
| | LP3 | 0.914 | 33.824 | | | |
| | LP4 | 0.951 | 54.443 | | | |
| | LP5 | 0.947 | 53.368 | | | |
| NSB pain | NSBP1 | 0.887 | 38.48 | 0.936 | 0.831 | 0.898 |
| | NSBP2 | 0.912 | 42.264 | | | |
| | NSBP3 | 0.935 | 63.36 | | | |
| Visual discomfort | VD1 | 0.768 | 14.077 | 0.932 | 0.697 | 0.913 |
| | VD2 | 0.906 | 61.562 | | | |
| | VD3 | 0.855 | 25.712 | | | |
| | VD4 | 0.812 | 19.406 | | | |
| | VD5 | 0.830 | 24.87 | | | |
| | VD6 | 0.833 | 21.082 | | | |
| Immersion | Im1 | 0.902 | 51.881 | 0.941 | 0.798 | 0.915 |
| | Im2 | 0.923 | 93.118 | | | |
| | Im3 | 0.912 | 63.65 | | | |
| | Im4 | 0.835 | 25.226 | | | |
| Interactivity | agecy1 | 0.919 | 68.595 | 0.961 | 0.861 | 0.946 |
| | agecy2 | 0.915 | 57.054 | | | |
| | agecy3 | 0.932 | 56.955 | | | |
| | agecy4 | 0.944 | 88.55 | | | |
| Intrinsic Motivation | Mov1 | 0.859 | 46.687 | 0.935 | 0.783 | 0.908 |
| | Mov2 | 0.892 | 41.937 | | | |
| | Mov3 | 0.905 | 46.968 | | | |
| | Mov4 | 0.884 | 40.145 | | | |
| Situational Interest | Int1 | 0.945 | 86.283 | 0.962 | 0.862 | 0.946 |
| | Int2 | 0.952 | 87.77 | | | |
| | Int3 | 0.946 | 70.395 | | | |
| | Int4 | 0.869 | 23.914 | | | |
| Embodied Learning | EL1 | 0.915 | 37.990 | 0.949 | 0.862 | 0.936 |
| | EL2 | 0.923 | 61.086 | | | |
| | EL3 | 0.948 | 79.640 | | | |
| self-efficacy | SE1 | 0.89 | 38.303 | 0.939 | 0.794 | 0.913 |
| | SE2 | 0.868 | 36.443 | | | |
| | SE3 | 0.911 | 51.172 | | | |
| | SE4 | 0.895 | 62.916 | | | |
| Extraneous Cognitive Load | CI 1 | 0.756 | 11.56 | 0.771 | 0.531 | 0.816 |
| | CI 2 | 0.628 | 5.518 | | | |
| | CI 3 | 0.792 | 10.293 | | | |

**Table 3. Discriminant validity.**

| | 1 | 2 | 3 | 4 | 5 | 6 | 7 | 8 | 9 | 10 | 11 |
|---|---|---|---|---|---|---|---|---|---|---|---|
| 1 | **0.924** | | | | | | | | | | |
| 2 | 0.867 | **0.934** | | | | | | | | | |
| 3 | 0.721 | 0.815 | **0.911** | | | | | | | | |
| 4 | 0.592 | 0.713 | 0.796 | **0.835** | | | | | | | |
| 5 | −0.28 | −0.288 | −0.217 | −0.302 | **0.894** | | | | | | |
| 6 | −0.302 | −0.309 | −0.295 | −0.365 | 0.754 | **0.928** | | | | | |
| 7 | −0.317 | −0.259 | −0.265 | −0.308 | 0.661 | 0.754 | **0.885** | | | | |
| 8 | −0.454 | −0.357 | −0.304 | −0.371 | 0.694 | 0.743 | 0.85 | **0.928** | | | |
| 9 | −0.314 | −0.238 | −0.238 | −0.338 | 0.69 | 0.725 | 0.817 | 0.848 | **0.93** | | |
| 10 | −0.262 | −0.172 | −0.15 | −0.255 | 0.705 | 0.713 | 0.662 | 0.749 | 0.756 | **0.891** | |
| 11 | 0.422 | 0.406 | 0.4 | 0.448 | −0.506 | −0.585 | −0.598 | −0.622 | −0.654 | −0.592 | **0.729** |

1: Head discomfort; 2: Limb pain; 3: NSB pain; 4: Visual discomfort; 5: Immersion; 6. Interactivity. 7: Intrinsic Motivation; 8: Situational Interest; 9: Embodied Learning; 10: Self-efficacy;11: Extraneous Cognitive Load; Note2: The square root of the AVE values shown in bold represent. Off-diagonal elements are the inter-construct correlations.

p < 0.001), supporting H1b, H1c and H1d. Interactivity demonstrated positive associations with intrinsic motivation (β = 0.701, t = 6.831, p < 0.001), situational interest (β = 0.553, t = 5.952, p < 0.001), embodied learning (β = 0.502, t = 4.239, p < 0.001) and self-efficacy (β = 0.41, t = 4.186, p < 0.001), supporting H2a, H2b, H2c and H2d. However, H2e was contradicted as interactivity reduced extraneous cognitive load (β = −0.566, t = 4.61, p < 0.001). Regarding physical discomfort, visual discomfort showed negative associations with immersion (β = −0.322, t = 2.838, p < 0.01) and interactivity (β = −0.347, t = 3.526, p < 0.001), supporting H3d and H4d. NSB pain demonstrated a positive association with immersion (β = 0.272, t = 2.040, p < 0.05), which was inconsistent with H3c.

## 4.4 Multi-group analysis

In order to analyze the moderating effects of different genders and ages, the entire sample was divided into two sub-samples according to individual categories (see Table 4). The first sub-sample was divided into two age groups according to the method adopted by Saredakis et al. [45]. This cutoff represents a theoretically meaningful midpoint, as prior research suggests that subtle age-related changes in vestibular and postural stability may begin to emerge around this period, which may in turn increase susceptibility to VR-induced physical discomfort [45]. 119 participants aged 35 years and older, and 90 participants aged below 35 years. The second sub-sample was divided into two groups by gender (94 males and 115 females). Multi-group PLS analysis, a technique that has been extensively adopted by previous studies [56], was performed to compare the differences between the path relationships in each sub-sample. The results of the multi-group analysis regarding gender showed that the women were more visually uncomfortable with VR, and that such visual discomfort affected the immersion and interaction in their VR experiences, hence the confirmation of H5d and H6d. Moreover, VR users below 35 years old were more sensitive to visual discomfort than VR users 35 years and older, and such visual discomfort affected the immersion during their VR experiences, hence the confirmation of H7d.

## 5 Discussions

This study examined how immersion and interactivity are associated with affective and cognitive learning constructs in VR-based nursing education, and how multiple forms of physical discomfort relate to these immersive and interactive experiences. The findings contribute to the understanding of VR-supported procedural training by clarifying the distinct roles of core VR features and by extending prior work on user discomfort into the domain of learning-related cognition.

                                                                    

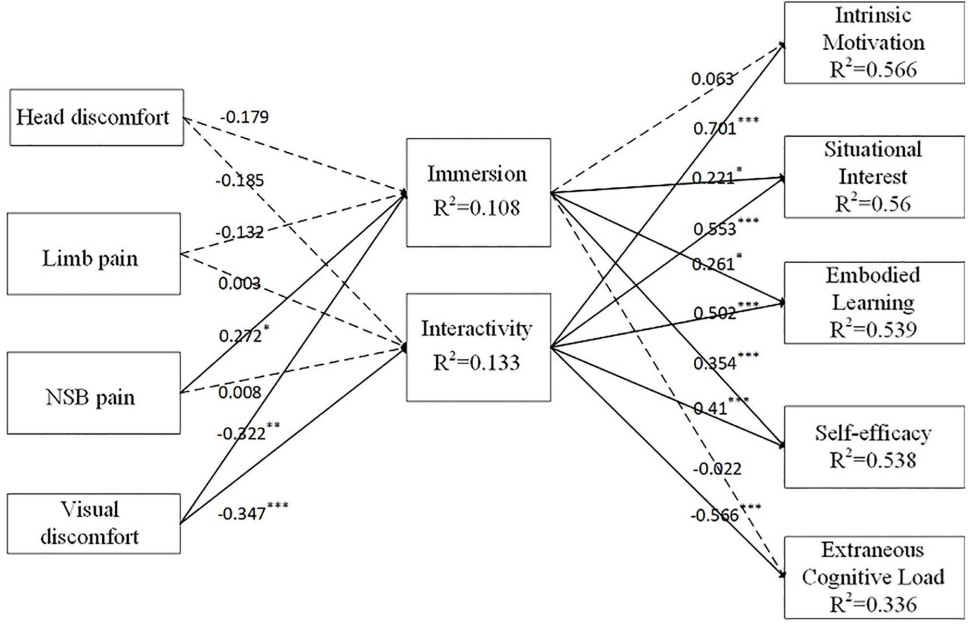

* P < 0.05; ** P < 0.01; *** P < 0.001
— Solid line: significant path
--- Dashed line: non-significant path

**Fig 2. Structural equation model illustrating the relationships among physical discomfort, immersion, interactivity, and learning-related outcomes in VR-based nursing training.** The model depicts the effects of head discomfort, limb pain, neck–shoulder–back (NSB) pain, and visual discomfort on immersion and interactivity, as well as the subsequent effects of immersion and interactivity on intrinsic motivation, situational interest, embodied learning, self-efficacy, and extraneous cognitive load. Standardized path coefficients are presented for all hypothesized paths, and the coefficient of determination ($R^2$) is shown for each endogenous construct. Solid lines indicate statistically significant paths, whereas dashed lines indicate non-significant paths.

The results showed that both immersion and interactivity were positively associated with intrinsic motivation, situational interest, embodied learning, and self-efficacy. These findings are consistent with previous studies indicating that immersive and interactive VR environments can enhance learner engagement, motivation, confidence, and conceptual understanding [3,5,6,57,58]. The present results also extend earlier work showing that interaction within VR supports action-oriented understanding [32], and that realistic contexts can strengthen self-efficacy by situating skills in meaningful practice environments [13]. A notable finding was that interactivity exhibited stronger associations with learning-related constructs than immersion. This pattern aligns with research suggesting that user agency, direct manipulation, and real-time feedback are particularly influential in procedural training scenarios that emphasize motor execution and task control [35]. At the same time, this study contrasts with Petersen et al. [32], who found that immersion, but not interactivity, was associated with situational interest in descriptive VR learning tasks. The inconsistency may stem from differences in task type: while descriptive learning emphasizes conceptual interpretation, procedural training requires active motor involvement, making interactivity a more central determinant of learning perceptions.

An additional unexpected finding was that interactivity was negatively associated with extraneous cognitive load, contradicting the initial hypothesis. Prior literature has suggested that interactive VR environments may increase cognitive load due to complex interfaces or simultaneous sensory demands [10,38]. However, in this study, interactivity was implemented through intuitive and low-complexity action pathways embedded within the procedural sequences. Such

**Table 4. Results of the multi-group analysis.**

**Group 1: < 35Y vs ≥ 35Y**

| | Path (<35Y) | Path (≥35Y) | Difference (<35Y - ≥35Y) | p-value | Significant difference? |
|---|---|---|---|---|---|
| HD -> Immersion | −0.047 | −0.305 | 0.258 | 0.639 | NO |
| HD -> Interactivity | −0.082 | −0.342 | 0.260 | 0.641 | NO |
| LP -> Immersion | −0.099 | −0.107 | 0.008 | 0.666 | NO |
| LP -> Interactivity | 0.078 | −0.047 | 0.125 | 0.689 | NO |
| NSBP -> Immersion | 0.130 | 0.328 | −0.198 | 0.495 | NO |
| NSBP -> Interactivity | −0.002 | 0.207 | −0.209 | 0.720 | NO |
| VD -> Immersion | −0.556 | −0.201 | −0.355 | 0.032 | YES |
| VD -> Interactivity | −0.488 | −0.235 | −0.253 | 0.270 | NO |

**Group 2: Female vs Male**

| | Path (Female) | Path (Male) | Difference (Female − Male) | p-value | Significant difference? |
|---|---|---|---|---|---|
| HD -> Immersion | −0.144 | −0.321 | 0.177 | 0.451 | NO |
| HD -> Interactivity | −0.101 | −0.281 | 0.180 | 0.739 | NO |
| LP -> Immersion | −0.072 | −0.113 | 0.041 | 0.956 | NO |
| LP -> Interactivity | 0.036 | 0.006 | 0.030 | 0.925 | NO |
| NSBP -> Immersion | 0.170 | 0.280 | −0.110 | 0.692 | NO |
| NSBP -> Interactivity | −0.023 | 0.181 | −0.204 | 0.463 | NO |
| VD -> Immersion | −0.333 | −0.101 | −0.232 | 0.037 | YES |
| VD -> Interactivity | −0.409 | −0.142 | −0.267 | 0.044 | YES |

HD: Head discomfort; LP: Limb pain; NSBP: NSB pain; VD: Visual discomfort.

design features may function as instructional scaffolding, guiding learners through the tasks and minimizing unnecessary processing demands, thus reducing extraneous load [59]. This finding underscores the importance of interface design in determining whether interactivity contributes to or alleviates cognitive burden.

Regarding physical discomfort, visual discomfort demonstrated the most consistent negative associations with immersion and interactivity. This finding aligns with prior evidence that head-mounted displays impose substantial visual burden, often resulting in eyestrain, dryness, and visual fatigue [23,60]. VR/AR review studies in healthcare similarly emphasize visual strain as a leading determinant of reduced user experience [27]. The present findings extend this body of work by showing that visual discomfort is not merely detrimental at the experiential level but is also meaningfully associated with reduced immersive and interactive learning perceptions in procedural VR contexts. By contrast, head discomfort, which is commonly manifested as dizziness or headache [44], did not show significant associations with immersion or interactivity, possibly because the present VR modules involved limited movement stimuli compared with high-motion VR applications such as driving or roller-coaster simulations.

A central and unexpected finding was the positive association between NSB pain and immersion. While earlier research has generally shown that musculoskeletal discomfort detracts from VR experience [41,42], two explanations may account for the current pattern. First, procedural nursing tasks require sustained concentration and repeated upper-body movements. Learners who were more engaged may have simultaneously reported higher immersion and greater musculoskeletal strain. In such cases, NSB pain may reflect the bodily load associated with high task engagement rather than contributing directly to immersive experience. Second, prior research suggests that sensorimotor involvement and bodily feedback can shape users' subjective experience in virtual environments [17]. From this perspective, musculoskeletal sensations may be integrated into the action–perception cycle during VR training, thereby co-occurring with heightened

immersion. These interpretations remain associative rather than causal, but they highlight the nuanced ways in which physical sensations may accompany immersive learning experiences in action-oriented VR tasks.

The multi-group analyses further revealed meaningful subgroup differences. Female participants exhibited stronger negative associations between visual discomfort and immersion/interactivity than males, which accords with previous findings that interpupillary distance and visual fatigue may disproportionately affect women during VR use [44]. Younger learners experienced stronger negative associations between visual discomfort and immersion compared with older participants. This pattern may relate to greater cumulative exposure to digital screens among younger adults, which has been linked to higher susceptibility to ocular strain [61]. These findings underscore the importance of considering individual differences when designing and implementing VR-based instruction.

Taken together, these results offer several practical implications. VR-based nursing education may benefit from incorporating adjustable hardware settings such as interpupillary distance, brightness control, and ergonomic headset configurations particularly for groups prone to visual discomfort. Shorter module durations or scheduled rest breaks may further reduce visual fatigue for younger or female learners [61,62]. Developers should continue to prioritize intuitive interaction design that supports task progression and minimizes extraneous load, and educators should select or design VR content with limited excessive motion to reduce the risk of discomfort.

Several limitations should be acknowledged. First, the study relied solely on self-reported measures, which may be subject to recall or response biases. Future research should incorporate objective indicators such as eye tracking, physiological measures, or behavioral performance data to provide a more comprehensive assessment of VR learning processes. Second, the cross-sectional design precludes causal inference; although PLS-SEM facilitates the testing of theoretically specified directional relationships, it is essential to clarify that SEM methodologies do not prove causal assumptions; rather, they lend credibility to them [62]. Future longitudinal or experimental designs are needed to more rigorously validate these mechanisms. Third, the findings may be specific to procedural VR tasks that require motor participation. Descriptive or observation-based VR learning tasks may elicit different discomfort profiles and interaction dynamics. Finally, the age cut-off used in the multi-group analysis follows prior VR literature but may not represent all developmental or experiential distinctions; future studies with larger and more diverse samples may better capture nonlinear age effects.

## 6 Conclusions

This study clarifies how immersion and interactivity are associated with key affective and cognitive learning constructs in VR-based nursing education, and shows that visual discomfort is a particularly influential factor that can diminish immersive and interactive engagement. Interactivity demonstrated a central role across learning-related outcomes, while subgroup differences indicated that younger and female learners may be more sensitive to visual strain. These findings offer practical implications for VR training design, including the use of adjustable headset settings, ergonomic configuration, intuitive interaction pathways, and shorter or structured sessions for users prone to visual fatigue. Overall, the results contribute to a more nuanced understanding of how VR technologies can be effectively integrated into nursing education while recognizing the physical and individual factors that shape learners' experiences.

## Supporting information

**S1 Appendix. VR training procedures and interaction design.** This appendix provides the full procedural steps, interaction modes (controller manipulation and gaze-based selection), standardized cueing system, and corrective feedback mechanisms used in the VR nursing training modules.
(DOCX)

**S2 Appendix. Questionnaire items used in the study.** This appendix contains all measurement items for immersion, interactivity, intrinsic motivation, situational interest, embodied learning, self-efficacy, extraneous cognitive load, and physical discomfort.
(DOCX)

## Author contributions

**Conceptualization:** Yu-Chia Chang, Cheng-chia Yang.

**Data curation:** Yi-Ting Lo.

**Formal analysis:** Yu-Chia Chang.

**Funding acquisition:** Yu-Chia Chang.

**Resources:** Yi-Ting Lo.

**Visualization:** Cheng-chia Yang.

**Writing – original draft:** Cheng-chia Yang.

**Writing – review & editing:** Cheng-chia Yang.

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
