## [Decision Letter · Decision Letter 0]

23 Nov 2025

Dear Dr. Yang,

Thank you for submitting your manuscript to PLOS ONE. After careful consideration, we feel that it has merit but does not fully meet PLOS ONE’s publication criteria as it currently stands. Therefore, we invite you to submit a revised version of the manuscript that addresses the points raised during the review process.

We look forward to receiving your revised manuscript.

Kind regards,

Muhammad Shahid Anwar

Academic Editor

PLOS ONE

Journal Requirements:

2. In the online submission form, you indicated that “The study was approved by the Institutional Review Board of Taichung Jen-Ai Hospital (Approval No. 110–82). According to the IRB’s ethical requirements, data are available upon reasonable request from qualified researchers who meet the criteria for access to confidential data. Requests for data access can be directed to the corresponding author (email: chengchia@asia.edu.tw) with approval from the Jen-Ai Hospital IRB.”

Reviewers' comments:

Reviewer's Responses to Questions

**Comments to the Author**

1. Is the manuscript technically sound, and do the data support the conclusions?

Reviewer #1: Yes

Reviewer #2: Yes

Reviewer #3: Yes

2. Has the statistical analysis been performed appropriately and rigorously?

Reviewer #1: Yes

Reviewer #2: Yes

Reviewer #3: Yes

3. Have the authors made all data underlying the findings in their manuscript fully available?

Reviewer #1: Yes

Reviewer #2: Yes

Reviewer #3: Yes

4. Is the manuscript presented in an intelligible fashion and written in standard English?

Reviewer #1: Yes

Reviewer #2: Yes

Reviewer #3: Yes

Reviewer #1: 1. The finding that neck–shoulder–back (NSB) pain increases immersion is unusual and contradicts most prior research. However, the manuscript only provides a brief speculative explanation.

2. The Methods section briefly states that the VR modules covered three nursing skills using Unity and Autodesk 3DS Max, but does not describe scenario difficulty, feedback design, or instructional cues.

3. provide more detailed descriptions of scenario complexity, interactivity levels, feedback mechanisms, and how consistency across participants was ensured. This information is necessary for reproducibility.

4. The paper employs PLS-SEM with a cross-sectional survey and a single VR session.

Clarify the justification for using PLS-SEM to test directional hypotheses, especially where causal language is used. Consider explicitly acknowledging limitations due to cross-sectional design and discuss how future longitudinal work could better validate the proposed causal mechanisms.

5. The manuscript lists construct names but does not present the exact questionnaire items in the main text; the reader is directed to “Appendix A,” which is missing in the PDF.

Include the full items or ensure Appendix A is attached, and clarify translation/validation procedures if the scales were adapted for a Taiwanese population.

Reviewer #2: Overall, the study shows that giving learners ways to actively interact with the system has a stronger impact than simply making the experience immersive. Interactivity boosts their confidence, motivation, and interest, and it also helps reduce unnecessary mental effort. The findings also show that when participants experienced visual discomfort—such as eye strain—it reduced both their sense of immersion and their willingness to interact. Interestingly, people who felt more neck, shoulder, or back pain actually reported feeling more immersed, possibly because they stayed in one posture for too long while deeply engaged. The study further notes that female participants tended to report more visual discomfort, which in turn lowered their immersion and interactivity, and younger participants were more sensitive to how visual discomfort affected their overall immersion.

Main Review Questions

(These are high-level questions that address the core methodology, interpretation, and contributions of the study.

1.On the Counter-Intuitive Finding of NSB Pain and Immersion:

The positive association between neck, shoulder, and back (NSB) pain and immersion (β = 0.272, p < 0.05) is a highly unexpected and central finding. The discussion posits this may be due to heightened bodily awareness enhancing presence. Could the authors elaborate on this interpretation and discuss alternative explanations? For instance, could this correlation be influenced by unmeasured factors, such as the duration or intensity of physical engagement required by the tasks that might independently induce both NSB pain and a higher sense of immersion, making the relationship spurious?

2.Reconciling Causal Hypotheses with a Cross-Sectional Design:

The study uses strong causal language in its hypotheses (e.g., "H1: Immersion with VR has positive impacts on...") and discussion, yet the cross-sectional design only allows for establishing correlations. A participant’s pre-existing high self-efficacy, for example, might cause them to perceive higher interactivity, rather than the other way around. How do the authors justify the causal framing of their findings, and could they explicitly address the limitations of inferring causality from their data in the discussion?

3.Investigating the Contradictory Finding on Interactivity and Cognitive Load:

The finding that interactivity significantly *reduced* extraneous cognitive load (β = -0.566) directly contradicts hypothesis H2e and the theoretical basis suggesting that complex interactions can increase it. While the authors attribute this to well-designed tasks, this is a significant finding with major implications for instructional design. Could the authors elaborate on how "interactivity" in this specific application was conceptually distinct from the learning task itself? Is it possible that the implemented interactivity functioned as a form of instructional scaffolding (thus reducing cognitive load) rather than as a separate, potentially distracting feature?

Minor Revision Questions

(These are more specific questions focused on clarity, completeness, and consistency, which are typically easier to address.)

1.Clarification of Exclusion Criteria:

Line 282 states that an exclusion criterion was "those who experienced discomfort during VR use." This appears to contradict the study's aim of measuring the effects of physical discomfort. Please clarify this criterion. Was it intended to exclude only participants who experienced debilitating discomfort that required them to withdraw from the training session?

2.Completeness of the Path Model in Figure 2:

In Figure 2, the path diagram only presents the final significant paths. For completeness and transparency, please consider adding the non-significant paths that were tested (e.g., from Head Discomfort and Limb Pain to Immersion/Interactivity) as dotted lines with their corresponding non-significant coefficients. This would provide a more comprehensive overview of the full model that was analyzed.

3.Rationale for the Age Cut-Off in Multi-Group Analysis:

The multi-group analysis uses a 35-year-old age cut-off, justified by citing Saredakis et al. [44]. Could the authors provide a brief rationale for why this specific cut-off is considered relevant for the current sample of nurse aide trainees? For instance, does this demarcation represent a meaningful split in terms of digital literacy, prior professional experience, or career stage within this cohort?

4.Operationalizing "Embodied Learning":

While Appendix A provides the questionnaire items for "Embodied Learning," the concept itself is complex. Could the authors add a brief operational definition in the Methods section (e.g., in 3.3 Measuring instrument) explaining what this construct is intended to capture in the specific context of procedural nursing skills in VR?

5.Specificity of Practical Implications:

The conclusions urge educators to "consider these factors when planning VR courses." To make the findings more actionable, could the authors offer more concrete suggestions? For example, based on the results, should training sessions for younger or female participants be designed to be shorter, include more frequent breaks, or utilize different hardware settings to mitigate the greater impact of visual discomfort?

6.The authors may consider citing and discussing the following related study, which presents a similar approach with your work. A. A. Laghari et al., “Quality of experience assessment in virtual/augmented reality serious games for healthcare: A systematic literature review,” Technology and Disability, vol. 36, no. 1–2, pp. 17–28, Feb. 2024, doi: 10.3233/tad-230035.

This paper could help strengthen the related work section and provide a clearer comparison with the proposed method.

Reviewer #3: The manuscript is overly lengthy and would benefit from tighter organization—several sections contain redundant explanations that hinder readability

Some hypotheses lack strong theoretical justification, especially those concerning physical discomfort; more prior literature should be cited to support these claims

The discussion of unexpected findings (e.g., NSB pain increasing immersion) is speculative and needs stronger evidence or alternative explanations

The multi-group analysis results are interesting, but the interpretation feels superficial—more depth is needed to explain gender and age differences

The study relies solely on self-reported questionnaires; the lack of objective physiological or behavioral measures limits the strength of the conclusions

**Do you want your identity to be public for this peer review?** For information about this choice, including consent withdrawal, please see our Privacy Policy

Reviewer #1: **Yes:** Dr. Shilpa Sharma

Reviewer #2: **Yes:** Avichandra

Reviewer #3: No

---

## [Author Response · Author response to Decision Letter 1]

30 Dec 2025

We sincerely thank the Editor and all three reviewers for their careful evaluation of our manuscript and for the constructive suggestions provided. We have revised the manuscript accordingly and believe that the revisions have substantially strengthened the clarity, rigor, and transparency of the study. Below, we provide point-by-point responses to each comment. All changes referenced below correspond to the revised manuscript.

Reviewer #1:

1. The finding that neck–shoulder–back (NSB) pain increases immersion is unusual and contradicts most prior research. However, the manuscript only provides a brief speculative explanation.

Response:

Thank you for pointing this out. In response, we have expanded the Discussion section to provide additional interpretation and to emphasize the associative nature of this finding. The revised text is provided below and has been added to the manuscript (Lines 436-448).

“A central and unexpected finding was the positive association between NSB pain and immersion. While earlier research has generally shown that musculoskeletal discomfort detracts from VR experience [60, 41], two explanations may account for the current pattern. First, procedural nursing tasks require sustained concentration and repeated upper-body movements. Learners who were more engaged may have simultaneously reported higher immersion and greater musculoskeletal strain. In such cases, NSB pain may reflect the bodily load associated with high task engagement rather than contributing directly to immersive experience. Second, prior research suggests that sensorimotor involvement and bodily feedback can shape users’ subjective experience in virtual environments [17]. From this perspective, musculoskeletal sensations may be integrated into the action–perception cycle during VR training, thereby co-occurring with heightened immersion. These interpretations remain associative rather than causal, but they highlight the nuanced ways in which physical sensations may accompany immersive learning experiences in action-oriented VR tasks.”

2. The Methods section briefly states that the VR modules covered three nursing skills using Unity and Autodesk 3DS Max, but does not describe scenario difficulty, feedback design, or instructional cues.

Response:

Thank you for this valuable comment. In response, we have expanded the Methods section to provide a concise description of how scenario complexity was controlled, how feedback was delivered, and how instructional cues were standardized across the three VR nursing modules. To maintain clarity in the main text, detailed procedural steps, interaction rules, and cueing and feedback specifications are provided in Appendix A. The revised text inserted into the Methods section is shown below (Lines 293-298).

“To enhance reproducibility and maintain consistency across participants, each module employed a standardized sequence of actions, fixed scenario complexity, and a unified cueing system (e.g., on-screen prompts and color-change indicators). Immediate corrective feedback was provided when required procedural steps were missed or performed incorrectly. Full procedural steps, interaction rules, and cueing/feedback specifications are documented in Appendix A”

3. provide more detailed descriptions of scenario complexity, interactivity levels, feedback mechanisms, and how consistency across participants was ensured. This information is necessary for reproducibility.

Response:

Thank you for this suggestion. We have added a concise methodological description in the Methods section specifying standardized action sequences, fixed scenario complexity, unified cueing, and immediate corrective feedback. Full procedural steps and interaction rules are provided in Appendix A.

4. The paper employs PLS-SEM with a cross-sectional survey and a single VR session.

Clarify the justification for using PLS-SEM to test directional hypotheses, especially where causal language is used. Consider explicitly acknowledging limitations due to cross-sectional design and discuss how future longitudinal work could better validate the proposed causal mechanisms.

Response:

Thank you for requesting clarification regarding the use of PLS-SEM. PLS-SEM was selected because it is well suited for models involving multiple latent psychological constructs, does not require multivariate normality, and performs robustly with moderate sample sizes. Importantly, the specified path directions reflect theoretical expectations, not causal claims.

In response, we have revised all hypothesis statements and related descriptions to use predictive or associative terminology (e.g., “is associated with”) instead of causal language. We have also added explicit statements in the Limitations section clarifying that, given the cross-sectional design, the findings should be interpreted as theory-driven directional associations rather than empirically confirmed causal effects. The revised text appears in the manuscript (Lines 470-474).

“Second, the cross-sectional design precludes causal inference; although PLS-SEM facilitates the testing of theoretically specified directional relationships, it is essential to clarify that SEM methodologies do not prove causal assumptions; rather, they lend credibility to them [63]. Future longitudinal or experimental designs are needed to more rigorously validate these mechanisms.”

In addition, the Methods section (3.4 Statistical methods) was slightly refined to clarify the analytical procedure and to explicitly reference established PLS-SEM evaluation guidelines (Hair et al., 2017), without altering the analytical strategy. The revised text appears in the manuscript (Lines 324-330).

“The study used SmartPLS 4, partial least squares structural equation modeling. PLS-SEM was selected because it is well suited for models involving multiple latent constructs, does not require multivariate normality, and performs robustly with moderate sample sizes. These characteristics align with the analytical needs of the present study, which includes several psychological constructs measured with multi-item scales. Model evaluation followed established PLS-SEM guidelines [53], including assessments of measurement reliability, convergent validity, discriminant validity, and structural path estimates.”

5. The manuscript lists construct names but does not present the exact questionnaire items in the main text; the reader is directed to “Appendix A,” which is missing in the PDF. Include the full items or ensure Appendix A is attached, and clarify translation/validation procedures if the scales were adapted for a Taiwanese population.

Response:

We thank the reviewer for this important comment. Appendix A has been included and now contains the complete list of questionnaire items. In the revised manuscript, we added a new Appendix A to provide technical details of the VR system/modules. As a result, the questionnaire items have been moved to Appendix B, which now contains the complete list of items.

Because the study used established scales that required minor adaptation for the VR nursing context, we followed a standard translation and modification procedure. Items were translated into Chinese, checked by two bilingual researchers for equivalence, and reviewed by three experts in nursing education and VR training to confirm relevance and clarity before finalizing the wording. As referred to in the following paragraph in the manuscript:(See Line 317-322).

“Because the scales were adapted for use in a VR nursing-training context, all items underwent a structured translation process. Items were first translated into Chinese by a professional translation agency. Two bilingual researchers reviewed the wording for semantic equivalence and conceptual consistency. Three experts in nursing education and VR-based training assessed item relevance and contextual appropriateness, leading to minor refinements before finalization. Complete items are provided in Appendix B.”

Reviewer #2: Overall, the study shows that giving learners ways to actively interact with the system has a stronger impact than simply making the experience immersive. Interactivity boosts their confidence, motivation, and interest, and it also helps reduce unnecessary mental effort. The findings also show that when participants experienced visual discomfort—such as eye strain—it reduced both their sense of immersion and their willingness to interact. Interestingly, people who felt more neck, shoulder, or back pain actually reported feeling more immersed, possibly because they stayed in one posture for too long while deeply engaged. The study further notes that female participants tended to report more visual discomfort, which in turn lowered their immersion and interactivity, and younger participants were more sensitive to how visual discomfort affected their overall immersion.

We appreciate the reviewer’s thoughtful synthesis of the study’s key findings. In response, we have refined the manuscript to clarify theoretical interpretations, address causal limitations, and enhance the methodological clarity and practical implications of the study.

Main Review Questions

(These are high-level questions that address the core methodology, interpretation, and contributions of the study.

1.On the Counter-Intuitive Finding of NSB Pain and Immersion:

The positive association between neck, shoulder, and back (NSB) pain and immersion (β = 0.272, p < 0.05) is a highly unexpected and central finding. The discussion posits this may be due to heightened bodily awareness enhancing presence. Could the authors elaborate on this interpretation and discuss alternative explanations? For instance, could this correlation be influenced by unmeasured factors, such as the duration or intensity of physical engagement required by the tasks that might independently induce both NSB pain and a higher sense of immersion, making the relationship spurious?

Response:

Thank you for raising this important point. We appreciate the opportunity to clarify this unexpected finding and have substantially expanded the Discussion section to provide a more comprehensive interpretation. As referred to in the following paragraph in the manuscript:(See Line 436-448).

“A central and unexpected finding was the positive association between NSB pain and immersion. While earlier research has generally shown that musculoskeletal discomfort detracts from VR experience [60, 41], two explanations may account for the current pattern. First, procedural nursing tasks require sustained concentration and repeated upper-body movements. Learners who were more engaged may have simultaneously reported higher immersion and greater musculoskeletal strain. In such cases, NSB pain may reflect the bodily load associated with high task engagement rather than contributing directly to immersive experience. Second, prior research suggests that sensorimotor involvement and bodily feedback can shape users’ subjective experience in virtual environments [17]. From this perspective, musculoskeletal sensations may be integrated into the action–perception cycle during VR training, thereby co-occurring with heightened immersion. These interpretations remain associative rather than causal, but they highlight the nuanced ways in which physical sensations may accompany immersive learning experiences in action-oriented VR tasks.”

2.Reconciling Causal Hypotheses with a Cross-Sectional Design:

The study uses strong causal language in its hypotheses (e.g., "H1: Immersion with VR has positive impacts on...") and discussion, yet the cross-sectional design only allows for establishing correlations. A participant’s pre-existing high self-efficacy, for example, might cause them to perceive higher interactivity, rather than the other way around. How do the authors justify the causal framing of their findings, and could they explicitly address the limitations of inferring causality from their data in the discussion?

Response:

We appreciate this important clarification. We agree that the cross-sectional design cannot establish causality or rule out reverse associations (e.g., learners with higher pre-existing self-efficacy perceiving greater interactivity). In response, we have revised all hypotheses and interpretations to adopt associative language and added an explicit limitation noting that causal inference is not possible with the present design. The revised text appears in the manuscript (Lines 470–474):

“Second, the cross-sectional design precludes causal inference; although PLS-SEM facilitates the testing of theoretically specified directional relationships, it is essential to clarify that SEM methodologies do not prove causal assumptions; rather, they lend credibility to them [63]. Future longitudinal or experimental designs are needed to more rigorously validate these mechanisms.”

3.Investigating the Contradictory Finding on Interactivity and Cognitive Load: The finding that interactivity significantly *reduced* extraneous cognitive load (β = -0.566) directly contradicts hypothesis H2e and the theoretical basis suggesting that complex interactions can increase it. While the authors attribute this to well-designed tasks, this is a significant finding with major implications for instructional design. Could the authors elaborate on how "interactivity" in this specific application was conceptually distinct from the learning task itself? Is it possible that the implemented interactivity functioned as a form of instructional scaffolding (thus reducing cognitive load) rather than as a separate, potentially distracting feature?

Response:

Thank you for this insightful comment. We have revised the Discussion section to clarify why interactivity in our VR system was associated with reduced extraneous cognitive load. Specifically, we now explain that interactivity in this study was implemented as intuitive, low-complexity action pathways (controller manipulation and gaze-based selection) that were tightly integrated with instructional scaffolding, rather than introduced as additional task demands. The revised text appears in the manuscript (Lines418–423):

“…However, in this study, interactivity was implemented through intuitive and low-complexity action pathways embedded within the procedural sequences. Such design features may function as instructional scaffolding, guiding learners through the tasks and minimizing unnecessary processing demands, thus reducing extraneous load [59]. This finding underscores the importance of interface design in determining whether interactivity contributes to or alleviates cognitive burden.”

Minor Revision Questions

(These are more specific questions focused on clarity, completeness, and consistency, which are typically easier to address.)

1.Clarification of Exclusion Criteria:

Line 282 states that an exclusion criterion was "those who experienced discomfort during VR use." This appears to contradict the study's aim of measuring the effects of physical discomfort. Please clarify this criterion. Was it intended to exclude only participants who experienced debilitating discomfort that required them to withdraw from the training session?

Response:

Thank you for pointing out this ambiguity. We have revised the manuscript to reflect this clarification (Lines 269–272):

“The exclusion criteria included individuals with neuropsychiatric, cardiovascular, cognitive, and sensory disorders, as well as those who experienced severe or debilitating discomfort during VR use that prevented them from completing the session”

2.Completeness of the Path Model in Figure 2:

In Figure 2, the path diagram only presents the final significant paths. For completeness and transparency, please consider adding the non-significant paths that were tested (e.g., from Head Discomfort and Limb Pain to Immersion/Interactivity) as dotted lines with their corresponding non-significant coefficients. This would provide a more comprehensive overview of the full model that was analyzed.

Response:

Thank you for this helpful suggestion. In the revised manuscript, Figure 2 now clearly displays both significant and non-significant paths, with non-significant paths represented by dashed lines and accompanied by their corr

---

## [Decision Letter · Decision Letter 1]

24 Feb 2026

Effects of Interactivity, Immersion, and Physical Discomfort on Learning in VR Nursing Education

PONE-D-25-54729R1

Dear Dr. Yang,

We’re pleased to inform you that your manuscript has been judged scientifically suitable for publication and will be formally accepted for publication once it meets all outstanding technical requirements.

Kind regards,

Muhammad Shahid Anwar

Academic Editor

PLOS One

Additional Editor Comments (optional):

Reviewers' comments:

Reviewer's Responses to Questions

**Comments to the Author**

Reviewer #1: All comments have been addressed

Reviewer #2: All comments have been addressed

2. Is the manuscript technically sound, and do the data support the conclusions?

Reviewer #1: Yes

Reviewer #2: Yes

3. Has the statistical analysis been performed appropriately and rigorously?

Reviewer #1: Yes

Reviewer #2: Yes

4. Have the authors made all data underlying the findings in their manuscript fully available?

Reviewer #1: Yes

Reviewer #2: Yes

5. Is the manuscript presented in an intelligible fashion and written in standard English?

Reviewer #1: Yes

Reviewer #2: Yes

Reviewer #1: (No Response)

Reviewer #2: The manuscript presents a comprehensive and well-executed investigation into the effects of immersion and interactivity in VR-based nursing education, with particular attention to learners’ affective, cognitive, and physical experiences. The study is well motivated, employs appropriate VR system design and robust analytical methods (PLS-SEM), and provides clear, meaningful results supported by adequate sample size and multi-group analysis. The findings offer important theoretical contributions and practical implications for designing effective and user-centered VR learning environments in healthcare education. Overall, the paper is methodologically sound, clearly written, and makes a valuable contribution to the field, and I recommend it for acceptance.

**Do you want your identity to be public for this peer review?** For information about this choice, including consent withdrawal, please see our Privacy Policy

Reviewer #1: **Yes:** SHILPA SHARMA

Reviewer #2: **Yes:** Avichandra Singh Ningthoujam

---

## [Editor Report · Acceptance letter]

PONE-D-25-54729R1

PLOS One

Dear Dr. Yang,

I'm pleased to inform you that your manuscript has been deemed suitable for publication in PLOS One. Congratulations! Your manuscript is now being handed over to our production team.

Kind regards,

on behalf of

Professor Muhammad Shahid Anwar

Academic Editor

PLOS One